# A Comparative Study on Microstructural Characterization of Thick High Strength Low Alloy Steel Weld by Arc Welding and Laser Welding

**DOI:** 10.3390/ma16062212

**Published:** 2023-03-09

**Authors:** Yunxia Chen, Xiao Xu, Yanjing Liu, Haichao Cui

**Affiliations:** 1School of Mechanical Engineering, Shanghai Dianji University, Shanghai 201306, China; 2Shanghai Key Laboratory of Materials Laser Processing and Modification, Shanghai Jiao Tong University, Shanghai 200240, China

**Keywords:** HSLA steel, microstructural characterization, laser welding, thick plate

## Abstract

Welding and the behavior of the weldments are important, since welding of high strength low alloy (HSLA) steels is a conventional method for manufacturing industrial parts. This work conducts a comparative investigation of microstructural characteristics and mechanical properties for joints of 16-mm-thick HSLA Q890 steel produced by multi-layer multi-pass shielded metal arc welding (SMAW) with filler wire and single-layer autogenous laser beam welding (LBW). The mechanical properties of the welded joints were assessed in terms of tensile and impact using butt joints. The results show that tensile failure occurred in the base metal during the tensile tests for most of the trials. The ultimate tensile strength and percent elongation of the LBW welded joint (973.5 MPa and 10%) are higher than those of the SMAW joint (951 MPa and 2.9%) due to the filler filling process of the SMAW process. The Charpy impact energy of the weld metal (16.4 J and 15.1 J) is lower than that of the heat-affected zone (18.5 J and 19.5 J) in the LBW joint and the SMAW joint.

## 1. Introduction

Welded structures of thick plate are broadly applied to pipelines to transmit gas and oil in large ships, offshore oil drilling platforms, pressure vessels, shipbuilding [1,2], bridges and storage tanks, railway components, nuclear power plants, pressure, and naval vessels [3,4]. HSLA steels have gained widespread use because they exhibit higher strength and better weldability than conventional high alloy steels [5].

The weldability of HSLA steel is reasonably good due to its reduced carbon equivalent [6]. The equipment used for these processes is quite large, and for medium and heavy plate thicknesses, workpieces can only be jointed piece by piece using fusion welding processes, such as submerged arc welding (SAW) [7,8], gas tungsten arc welding (GTAW), gas metal arc welding (GMAW) [9,10], and SMAW [11]. Arc welding has productivity limits due to its low penetration depths. Because of this, multi-pass welding is required for these thick plate structures, which are cut with large-sized single V grooves and require multi-layer multi-pass arc welding, potentially presenting some disadvantages, such as gas porosity, slag inclusion, large levels of distortion and residual stress, poor welding quality defects, and low productivity [12,13]. In addition, other welding processing methods of HSLA steels joints include electron beam welding (EBW), LBW, and friction stir welding (FSW). However, the size of parts for EBW is still a challenge because the weld environment needs to be set in a vacuum chamber, the FSW of thick plates is constrained due to the size of the rotating tool, and the required rigidity of the welding machine [14,15,16]. In contrast, LBW is particularly attractive for innovative and cost-effective applications because of its high precision, low heat input, narrow weld pool, small welding deformation, and the narrow width of the heat-affected zone (HAZ) [17,18]. Nevertheless, the application of LBW to HSLA steels has not been widely adopted for practical applications due to the short cooling times after welding [19,20,21].

Published comparative studies on SMAW and LBW of HSLA steel plates with higher thicknesses are very scant. The behavior of weldments has been related to microstructural features [22,23]. Therefore, studying the effect of different welding techniques and their parameters on thick Q890 HSLA steel is essential. The present study was carried out to compare the microstructural characteristics and mechanical properties of 16 mm thick Q890 HSLA steel weldment fabricated by SMAW and LBW techniques.

## 2. Experimental Details

The 16-mm-thick HSLA Q890 steel used in this investigation was provided by Baoshan Iron and Steel Co., Ltd., (Shanghai, China). The main chemical compositions (in wt.%) of the steel and X90 filling wire are given in Table 1. The equivalent carbon content (CE) for this steel grade amounted to about 0.55 according to SS-EN1011-2 [24], indicating a relatively high hardenability. Its parent microstructure was of a predominantly high temperature tempering structure of martensite (M). In addition, 200 mm × 200 mm welded blanks were fabricated by butt welding. LBW equipment consisted of an IPG Photonics 10 kW fiber laser system (IPG-10000, Burbach, Germany) mounted on a Fronius Transpuls Synergic 5000 wire feeding machine (Fronius, Pettenbach, Austria), and a system of shielding gas 99.99% pure Argon was used as the protective gas, which had a flow rate of 15 L/min. The laser beam was perpendicular to the steel sheet being welded. The weld geometries and dimensions are presented in Figure 1a,b. Prior to welding, mechanical polishing of the BM was performed and the oil on the BM surface was cleaned using acetone.

The LBW experiments were carried out with a laser power of 8 kW, a welding velocity of 0.3 m/min, and a negative defocusing length of 4 mm.

The welding process of SMAW consisted of six layers and twenty-one passes. The basic parameters of SMAW were set as: a welding current of 250 A; a welding velocity of 0.11 m/min; a welding voltage of 28 V. To prevent cracks, the preheating temperature before welding was 120 °C, the inter-pass temperature was strictly controlled below 200 °C, and post-weld heat treatments were adopted. After welding, the weldment was cooled in air for 20 min, heated to 360 °C in 60 min, and kept constant for 30 min in a heat treatment furnace before being cooled in air.

Nine replications of each level of two joints were made to achieve measurement accuracy. Out of those, three were selected for tensile tests, four were employed in impact tests, and two were used for microhardness and microstructure observations. Following GB/T 228-2002 [25], tensile test specimens were sectioned from the welded blanks, as presented in Figure 1a,b. Tensile tests were performed on a Zwick/Roell Z100 (Zwick Roell Group, Ulm, Germany) universal testing facility with a constant strain rate of 1 mm/min at room temperature. The tensile test recorded for each joint was the average of three trials. The schematic of tensile test specimens is presented in Figure 1c.

The Charpy V-notch impact tests were performed at −40 °C (according to GB/T 229-2007 [26]). Four different test sampling locations in each joint were designed in view of the effect of the sampling position on the low temperature impact property, as shown in Figure 1a,b. The absorbed energy value recorded for each joint was the average of four trials. The schematic of the unstandardized 2.5 × 10 × 55 mm specimen with a V-notch is presented in Figure 1d.

The metallographic specimens were prepared by cutting along the cross section of the weld followed by a conventional procedure, such as mounting, grinding, polishing, or etching. Thereafter, the etched samples were observed using OM (Leica DM4000 M LED, Leica Camera, Wetzlar, Germany). Microstructures of the unetched cross sections were characterized by using SEM (JEOL JSM-7800F Prime, JEOL Ltd., Tokyo, Japan) with EDX.

Vickers microhardness tests (Zwick/Roell ZHμ, Zwick Roell Group) were utilized at the mid-thickness of the weld cross section to evaluate the microhardness of the weld metal (WM) and HAZ. According to ASTM:E384, an indentation load of 500 g with a dwell time of 15 s was employed for each test.

## 3. Results and Discussion

### 3.1. LBW Weld Microstructure

The HAZ can be described by four subzones at a fine-grained level: the coarse-grained heat affected zone (CGHAZ); the fine-grained heat affected zone (FGHAZ); the two-phase zone (TPZ); and the over-tempering zone (OTZ) in detail. The definition of each zone is as follows: the fusion line is used as the boundary between the weld metal (WM) and HAZ; the boundary between the CGHAZ and the FGHAZ is about 10 μm in width with obvious grain size growth; the boundary between the FGHAZ and the TPZ is where the tempered sorbite of the base metal (BM) disappears; the boundary between the TPZ and the OTZ is based on the disappearance of white striped microstructures and the appearance of obvious granular carbides. Each zone width was calculated using the average width method that was equal to the zone area divided by its thickness using Image-Pro Plus software: IPP 7.0.

Figure 2a shows the optical morphology of the LBW joint. It can be seen that the weld morphology is good without any obvious defects. Although the carbon equivalent (0.55%) of the Q890 HSLA steel is a little higher, the LBW performance of the welded joint had been greatly improved by the preheating treatment before welding. It can also be seen that the maximum and minimum bead width of LBW are about 9.5 mm and 3.6 mm, respectively. The maximum and minimum bead width of the HAZ are 4 mm and 2.2 mm, respectively, and its average bead width is 2.9 mm. The weld penetration is about 13 mm. The LBW weld can be divided by ① WM, ② CGHAZ, ③ FGHAZ, ④ TPZ, ⑤ OTZ, and ⑥ BM.

① Weld Metal

In the LBW joint, the weld seam was formed by the solidification of partial base metal after high temperature melting. It was mainly composed of a certain amount of equiaxed crystal phases surrounded by coarse columnar crystal phases. The columnar crystal phases grew along the perpendicular direction to the side wall of the molten pool from the base metal to the weld seam, as shown in Figure 3. The width of columnar crystal phases is about 10 μm, and their maximum length can be greater than 200 μm. The mechanical properties of the joint significantly decreased because of the existence of the columnar crystal phases. The WM was mainly composed of ferrite (F) and bainite (B). The ferrite was mainly proeutectoid ferrite, which is an important component of columnar crystal phases, while the equiaxed crystal phases were almost entirely composed of polygonal ferrite phases. In addition, there were a few acicular ferrite phases and a certain amount of martensite phases. Acicular ferrite phases formed with some inclusions as the core and most martensite phases were distributed in the fusion line, at the root or top of the weld due to faster cooling rate.

② Coarse-Grained Heat-Affected Zone

The CGHAZ is adjacent to the WM, and its higher peak temperature of the thermal cycle ranges from 1100 °C to the liquidus temperature. The austenite phases can stay in this temperature range for a long time and dramatically grow, and then form larger overheating phases during cooling, which can worsen the mechanical performance of the whole joint. The average grain size of the CGHAZ can reach about 50 μm; even 100 μm for the grains near to the fusion line.

The microstructure features of the CGHAZ are closely related to the heat input. When the heat input increases, acicular ferrite begins to gradually form in the CGHAZ. Acicular ferrite with low hardness and good toughness can significantly improve the impact behavior of the CGHAZ. If the heat input keeps increasing and the cooling rate decreases, polygonal ferrite phases will increase and acicular ferrite will decrease. Meanwhile, there is a small amount of pearlite that can possibly appear in this area.

Due to the lower heat input and faster cooling rate, there are more and more strip-shape bainite, ferrite, and low-carbon martensite phases formed in the CGHAZ. Most of the strip-shape phases initially form from the original austenite grain boundary and grow toward the crystal at a certain angle; each strip maintains a certain phase relationship and a fine structure inside the strip. With lower heat input (9.6 kJ/cm) in the LBW, the internal phases in the CGHAZ are low-carbon martensite and a few bainite and ferrite, as shown in Figure 4.

③ Fine-Grained Heat Affected Zone

During the LBW process, for the FGHAZ, the peak temperature of the welding thermal cycle is between 1100 °C and A_C3_ temperature, and its mechanical properties performance is the best out of all the weld joints. The reason for this is that the microstructure in the FGHAZ undergoes a recrystallization and new grain refinement during heating and cooling; more fine grains can be obtained, which is equivalent to the base metal normalized. In general, most grain sizes of the fine grains are less than 10 μm. Observed under a high magnification microscope, the microstructure of the FGHAZ can be described by some finer martensite, bainite, and few fine carbides precipitated between ferrite blocks, as shown in Figure 5.

④ Two-Phase Zone

The range of the TPZ in the LBW joints is wider (especially in the center of the joint), and its peak temperature of the welding thermal cycle is between A_C3_ and A_C1_. During this temperature range, only partial base metal can obtain sufficient heat to finish phase transformation and form complex phases based on sorbite and the partial transformation structure.

It can be seen from Figure 6a,b that there are some obvious striped microstructures in the TPZ. The reason for these striped microstructures is due to interlacing ferrite and other phases along the deformation direction when heating. The appearance of striped microstructures makes the steel anisotropic, and has a negative effect on the mechanical properties of the steel. In the base metal, there are few striped microstructures can be less obviously seen. However, striped microstructures can be easily found after welding because the secondary cementite of the austenite with partial phase transformation in the TPZ precipitates along the original strip direction and forms duplex phases of ferrite and cementite. These striped microstructures can be seen more clearly due to their own fine structures, making most of the incomplete phase transformation turn black after corrosion.

The striped microstructures in the TPZ are shown in Figure 6c,d, under the scanning electron microscope. The black part is similar to the tempered sorbite duplex phase, which is mainly composed of striped ferrite, carbides, and few polygonal ferrite. However, there are more granular carbides among the ferrite phases, and black striped microstructures increase with the precipitation of carbides, which results in a finer microstructure and a stronger black color. The white part of the striped microstructures is mainly irregular tangled ferrite bands and a few precipitated carbides. Compared with the black part, this part looks white and its hardness is lower due to obvious ferrite characteristics; the reason is that when carbon-poor austenite reaches the austenitizing temperature, there are more proeutectoid ferrite precipitated, and then ferrite and cementite are precipitated, which results in the formation of some complex ferrite morphologies.

⑤ Over-Tempering Zone

Due to the OTZ being far away from the WM, its lower peak temperature of the welding thermal cycle is not more than A_C1_ and its cooling speed for welding is less. Therefore, the base metal in the over tempered zone does not undergo austenitizing transformation in the welding thermal cycle; it seems to be double tempered at higher temperature than that of the quenching and tempering. The higher temper temperature is beneficial for the carbide particles to aggregate and grow up in this area, which results in more formations of polygonal ferrite phases, which weaken the microhardness. Therefore, the OTZ has a significant responsibility for the softening of the welded joint.

From the discussion of the OTZ in the welded joint, it can be observed that there are more granular carbides dispersed on the ferrite matrix; these granular carbides probably originate from the sorbite phase carbides in the base metal, where they aggregate and grow up and are distributed on the ferrite grain boundary after double tempering, as shown in Figure 7. The area with more coarse granular carbides is black, while the polygonal ferrite phase is white. In the OTZ, there are more carbides precipitated, grown, and coarsened, which causes the hardness of the OTZ to decrease. Therefore, it is the most softened zone in the whole welded joint.

In order to figure out the phase constituents of the joint softening zone, EBSD was used and the results are shown in Figure 8. The EBSD results show that the Fe_3_C fraction of the phase constituents is 0.77% and much higher, as shown in Table 2. The reason is that there is no post-weld heat treatment after welding, which results in some residual stress in the welded joint; the lower resolution rate is 75.23%. The face-centered cubic (FCC) austenite fraction of the phase constituents is only 0.01%, which means that the amount of retained austenite in the laser-welded joint is very small, and the body-centered cubic (BCC) ferrite fraction of the phase constituents is 74.44%. According to the previous analysis, this zone consists of some polygonal ferrite and few sorbite phases. These polygonal ferrite phases are responsible for the hardness decreasing.

### 3.2. SMAW Weld Microstructure

Figure 2b shows the microstructure of the whole arc welded joint. It can be seen that 21 passes of SMAW were need to fill the V-groove for 16-mm thick weldment. The heat input was higher than 16.4 kJ/cm and the maximum bead width was approximately 30 mm. The joint morphology was good and the maximum bead width was about 30 mm; the average width of HAZ between the BM and the WM was nearly 2.5 mm; while the width of HAZ in the inner WM was only 0.5 mm. This can be explained by the fact that the HAZ is produced by reheating the previous part of the weld with the following weld in the area where the welds overlap with each other. The formation of HAZ leads to the change in microstructure, which in turn has an impact on the performance of the joint. Finally, the typical solidification structure, such as the columnar crystal and the equiaxed crystal, appears in each layer and in each pass; most of them are the column crystal.

The microstructure in the SMAW joint is almost similar to that in the LBW joint overall. However, if a higher heat input (16.4 kJ/cm) and slower cooling rate were used, the microstructure in the SMAW joint would show some differences that the LBW joint would not have. The following discussion focuses on the microstructure differences of the WM and HAZ.

In the multi-layer multi-pass SMAW, there are many HAZs that are heated many times in the WM, which results in obvious secondary HAZs existing among each layer and each pass of the WM, as shown in Figure 9a. The reason for this is similar to the formation of the black microstructures in the TPZ of the LBW HAZ. Due to the higher heat input and slower cooling rate, carbides in the secondary HAZ have sufficient time to precipitate and grow up among each layer of the WM, forming complex phases of ferrite and cementite with finer microstructures and a darker color. Figure 9c is a local magnification photo of Figure 9a. The typical secondary HAZ and WM can be easily seen in the 2-mm weld pass. The WM is mainly composed of equiaxed crystal and columnar crystal phases, and the equiaxed crystal phases are composed of some acicular and polygonal ferrite and bainite. The columnar crystal is mainly made up proeutectoid ferrite and bainite ferrite, as shown in Figure 9c. The size of the equiaxed crystal phase is around several microns, and the width of the columnar crystal phase is 150 μm; the width of the black secondary HAZ is almost 250 μm. Typical bainite ferrite clusters can be seen in the equiaxed crystal phase in Figure 10b; the length of ferrite is 70 μm and the width is 3 μm.

The HAZ of the SMAW is also composed of a CGHAZ, a FGHAZ, a TPZ, and an OTZ. The grain size of the CGHAZ is nearly 100 μm and consists of coarse martensite and bainite. Because of the higher heat input and slower cooling rate, most phases of the CGHAZ are bainite with a few polygonal ferrite. In Figure 10, the grain size in the FGHAZ is nearly 10 μm. The TPZ consists of some duplex microstructures and ferrite phases. The more the HAZ is away from the WM, the lower the decrease in the peak temperature and cooling rate of the thermal cycle, and the less carbide precipitate. The precipitated carbide is discontinuously distributed and becomes a tempered sorbite phase when it reaches the TPZ. In view of the post-weld heat treatment, it is not easy to find black and white striped phases of the TPZ in SMAW.

The total heat input of the SMAW with filler wire was greater than that of the LBW; the difference in microstructure resulted in different performances of their mechanical properties. The comparison results are shown in Table 3.

Meanwhile, the differences in microstructure can be described through the content of phases and the width of each zone in particular. In order to find the difference between each welding, the abovementioned average statistical method was used and the statistical results are shown in Figure 11.

The results show that significant differences between two joints are as follows:

The joint weld by multi-layer multi-pass SMAW results in more complex microstructures and double-HAZ, which are composed of some black duplex phases, like those of the two-phase zone, and there are much more proeutectoid ferrite phases in the WM. The average bead widths of the LBW joint and the SMAW joint are 4.2 mm and 15.4 mm, respectively. The average bead width of the LBW is significantly smaller than that of the SMAW. The performance of mechanical properties in the WM is always poorest in the whole joint. The decrease in bead width in LBW is helpful in order to improve the performance of the whole welded joint.

There is minor difference in the width of the CGHAZ and the FGHAZ for the two joints. The difference is that there is more bainite phases in the CGHAZ and the FGHAZ in the SMAW joint, and several chained carbide particles along the grain boundary; the most probably reason for this is the larger heat input used.

The TPZ is not obvious and its width is 0.8 mm in the SMAW joint, while the width in the LBW joint can reach 1.5 mm, or even more than 2.5 mm, and the performance of the TPZ is slightly lower than that of the base metal. Therefore, the performance of the LBW joint could worsen due to its wider TPZ.

The width of the OTZ in the two joints are almost same and equal to 0.4 mm. The microstructure has no obvious change and it has improved to a degree because of the overheating treatment in the SMAW. However, it is not good for the performance of the OTZ to improve the LBW without any post-weld heat treatment.

### 3.3. Mechanical Properties

Microhardness test results

In Figure 12a, the microhardness variations across the middle of the SMAW joint are presented. The hardness of the BM is about 360 HV_0.3_, which is similar to that of the LBW, indicating that post-weld heat treatment had little effect on the hardness of the BM. Because of multi-layer and multi-pass welding, the transverse hardness fluctuation of the arc-welded joint is obvious. It can be seen that the whole hardness spot experienced four weld bead ups and downs, and each weld bead had some degree of hardening and softening, but due to the effect of post-weld heat treatment, the hardening and softening effect was not obvious. The highest hardness is 430 HV_0.3_ in the HAZ between each pass, which is about 30 HV_0.3_ lower than that of the LBW joint, which is the result of the post-weld heat treatment.

Figure 12b depicts the Vickers microhardness variations across the middle of the LBW joint. The hardness of the BM is about 360 HV_0.22_, and the WM are formed by the remelting of the BM. Because the WM had not undergone any post-weld heat treatment, its hardness is higher than that of the BM, which is about 400 HV_0.22_. There is a certain degree of softening in the OTZ; the lowest hardness is only 334 HV_0.22_, and the width of the whole softening zone is about 1.2 mm.

Tensile test results

The stress–strain curves for the LBW and SMAW joints are shown in Figure 13, and the average values of the tensile tests for the LBW and SMAW joints are summarized in Table 4. The average yield strength *R*_p0.2_, the average ultimate tensile strength *R*_m_, and the average elongation *A* of LBW joints are 898 MPa, 973.5 MPa, and 10%, respectively. Compared to those of the LBW joints, the decline in the average yield strength (896 MPa) and the average ultimate tensile strength (951 MPa) in the SMAW joints is not glaring, except that the average elongation (2.9%) dramatically decreases. It could be that increasingly more weld microstructures are involved in the tensile behavior from the root of the weld to the top of the SMAW joint, resulting in a smaller elongation of the tensile coupon. In Table 4, it can be found that the fracture of most of specimens occurred in the BM for the LBW and SMAW joints; only one fracture occurred in the WM of LBW joint due to weld defects.

Impact test results

Table 5 depicts results of the impact tests at −40 °C, showing that the Charpy impact energy of the WM (16.4 J) is less than the HAZ (18.5 J) samples employed in the LBW joints. Additionally, the Charpy impact energy of the WM (15.1 J) is significantly less than the HAZ (19.5 J) for all SMAW samples. The Charpy impact energy of the WM for the SMAW samples is lower than that of the LBW samples. The presented post-weld heat treatment in the SMAW process could influence part of the HAZ and improve HAZ toughness. Because of this, the Charpy impact energy of the HAZ for the SMAW samples is a little higher than that of the LBW samples.

Additionally, the prescribed table values came from the unstandardized impact specimens (2.5 × 10 × 55 mm) compared to ISO standard samples (10 × 10 × 55 mm); an equivalent Charpy impact energy value is equal to the presented value multiplies 2 or 4 [27]. Therefore, even though the minimum value of the WM for the SMAW sample is equivalent to 26.92 J, it meets the requirement for the Charpy impact energy value (27 J).

Figure 14 shows the impact fracture photo of the WM of the LBW joint. From Figure 14a, it can be seen that there is almost no fiber area, the fracture surface is relatively flat, and pits occasionally exist, showing brittle fracture characteristics. Figure 14b is an enlarged view of the fracture fiber area. It can be seen that there are relatively few dimples and a pore with a size of 10 μm. Figure 14c is the morphology of the fiber region at higher magnification, and dimples vary in size between 0.5–3 μm. Figure 14d shows a high-magnification dimple photo in the radiative region, where the shear characteristics are more significant compared with Figure 14c. On the whole, the fracture impact energy of the WM is low, with 11.50 J, and the mode of failure is brittle.

## 4. Conclusions

In the present work, the mechanical properties and microstructural features of SMAW and LBW welded joints for 16-mm-thick HSLA Q890 steel were discussed. The microhardness of the welded joint results showed that the LBW was slightly higher than the SMAW, meaning that the post-weld heat treatment associated with the SMAW process can improve microhardness. Furthermore, the tensile results showed that tensile failure occurred in the base metal; the ultimate tensile strength and percent elongation of the LBW welded joint (973.5 MPa and 10%) were higher than those of the SMAW joint (951 MPa and 2.9%) due to the filler filling of the SMAW process. Finally, the impact results showed that the Charpy impact energy of the weld metal (16.4 J and 15.1 J) was lower than that of the heat-affected zone (18.5 J and 19.5 J) in the LBW joint and the SMAW joint.

## Figures and Tables

**Figure 1 materials-16-02212-f001:**
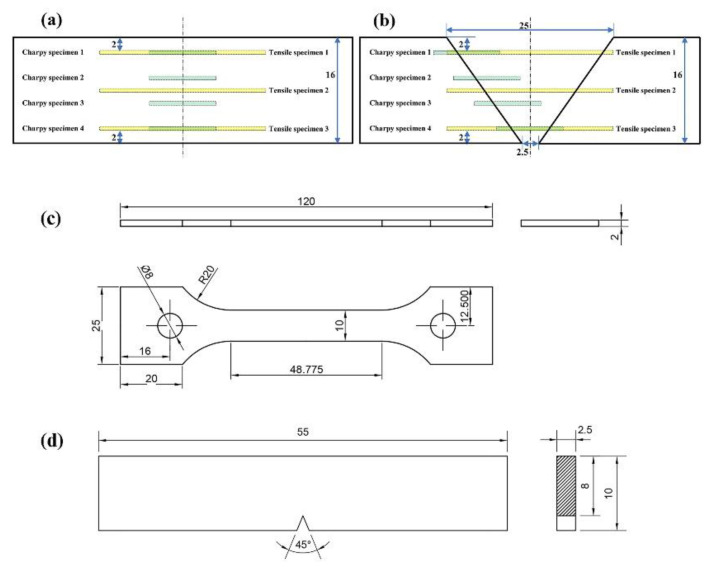
Dimensions of grooves shape (**a**) LBW joint (**b**) SMAW joint (**c**) Tensile specimen (**d**) Charpy specimen (Unit: mm).

**Figure 2 materials-16-02212-f002:**
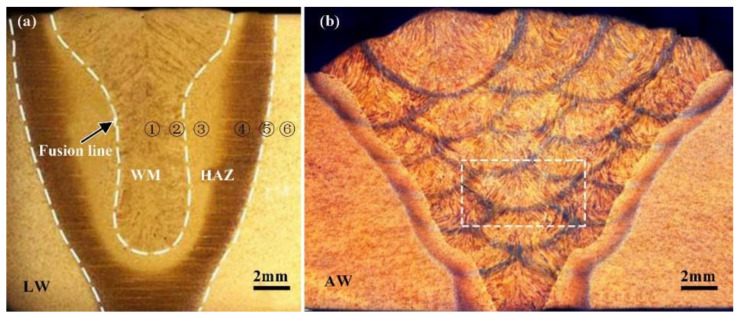
Optical morphology of joints (**a**) the LBW (**b**) the SMAW.

**Figure 3 materials-16-02212-f003:**
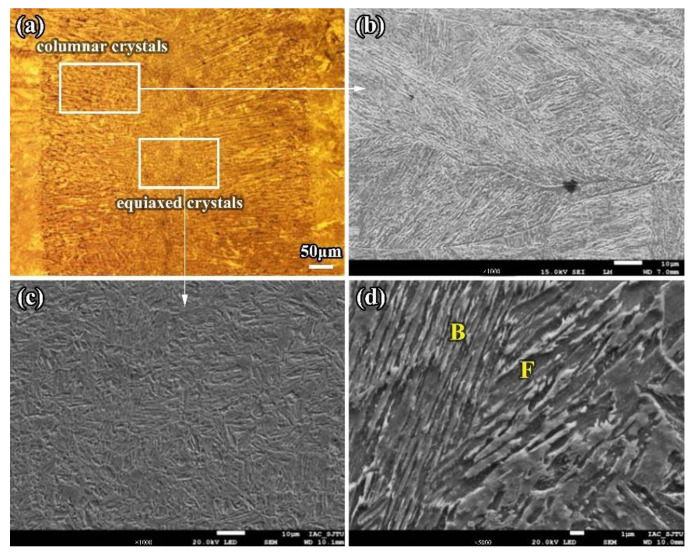
Microstructure of WM of the LBW joint (**a**) Macro-photo in weld center area (**b**) Columnar crystal (**c**) Equiaxed crystal (**d**) Ferrite (F) and bainite (B).

**Figure 4 materials-16-02212-f004:**
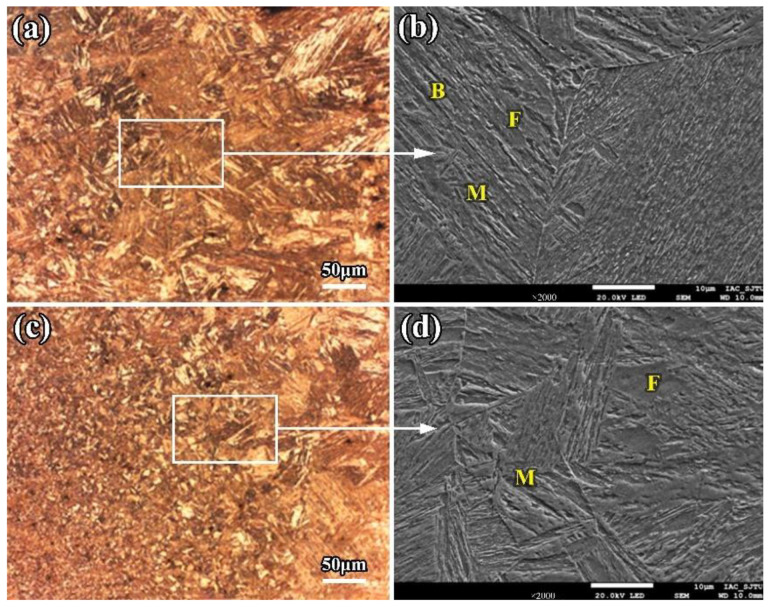
Microstructure of the CGHAZ of the LBW joint (**a**) CGHAZ close to fusion line (**b**) Martensite and bainite (**c**) CGHAZ far from fusion line (**d**) Martensite and ferrite. F: ferrite; M: martensite; B: bainite.

**Figure 5 materials-16-02212-f005:**
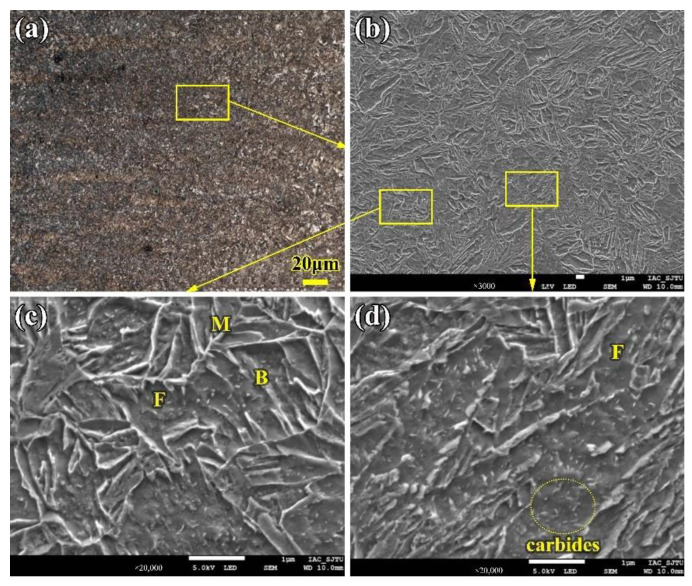
Microstructure of the FGHAZ of the LBW joint (**a**) Macro-photo of the FGHAZ (**b**) Micro-photo of the FGHAZ (**c**) Ferrite and bainite (**d**) Carbides in the FGHAZ. F: ferrite; M: martensite; B: bainite.

**Figure 6 materials-16-02212-f006:**
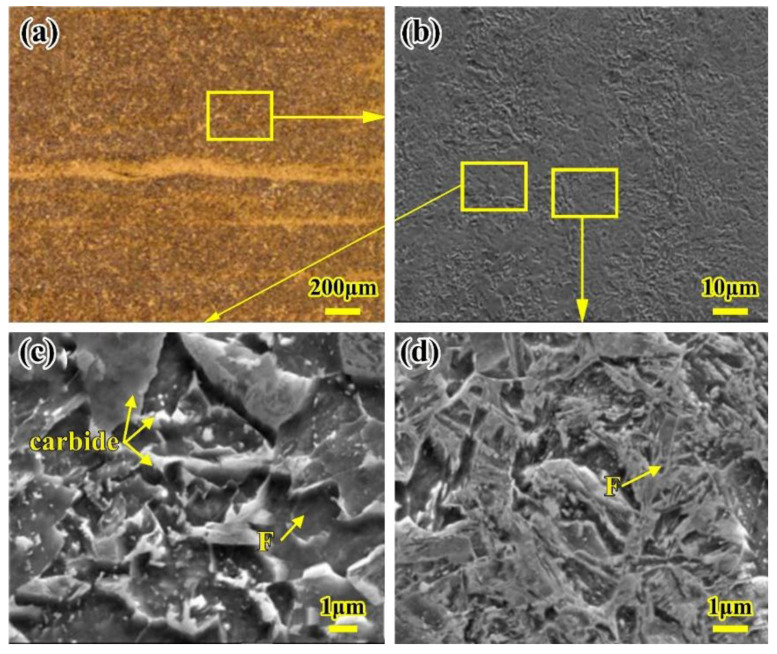
Microstructure of the TPZ of the LBW joint (**a**) Segregation band in the TPZ (**b**) Microstructure of the segregation band in SEM (**c**) Microstructure in black part (**d**) Microstructure in white part. F: ferrite.

**Figure 7 materials-16-02212-f007:**
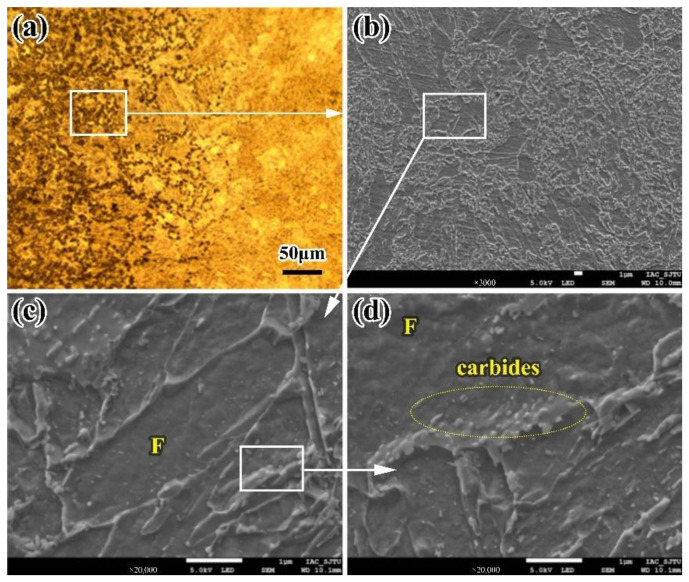
Microstructure of the OTZ of the LBW joint (**a**) Microstructure in the OTZ (**b**) SEM in the OTZ (**c**) Polygonal ferrite (**d**) Carbides on ferrite grain boundaries. F: ferrite.

**Figure 8 materials-16-02212-f008:**
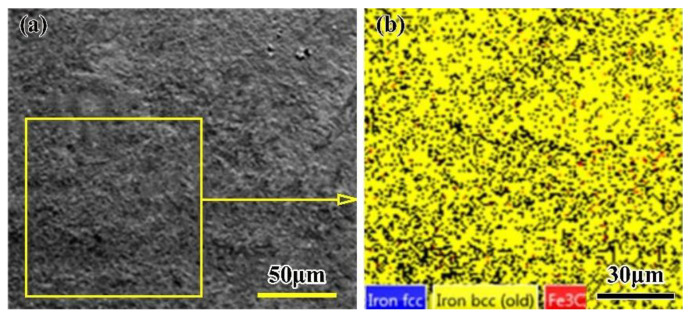
Results of EBSD in the OTZ of the LBW joint (**a**) Location of EBSD (**b**) Phase distribution.

**Figure 9 materials-16-02212-f009:**
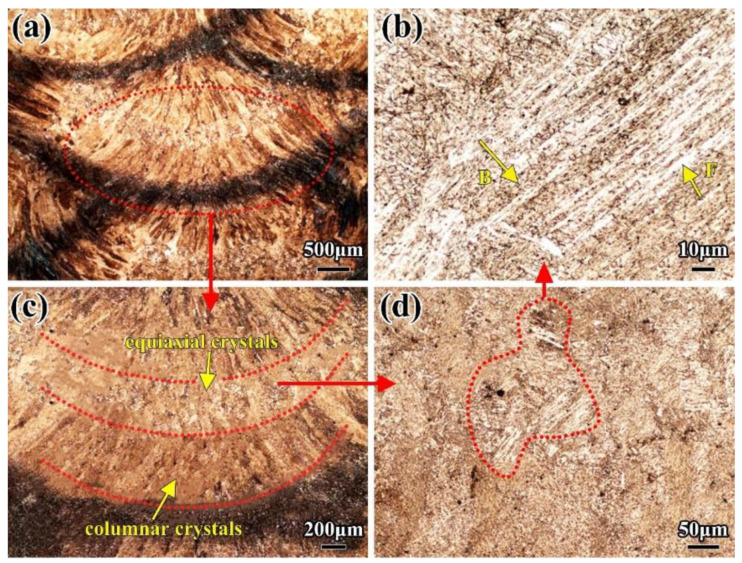
Microstructure of SMAW joint (**a**) Macro-photo in weld metal (**b**) Bainite ferrite in weld metal (**c**) Equiaxial and columnar crystals (**d**) Equiaxial crystals. F: ferrite; B: Bainite.

**Figure 10 materials-16-02212-f010:**
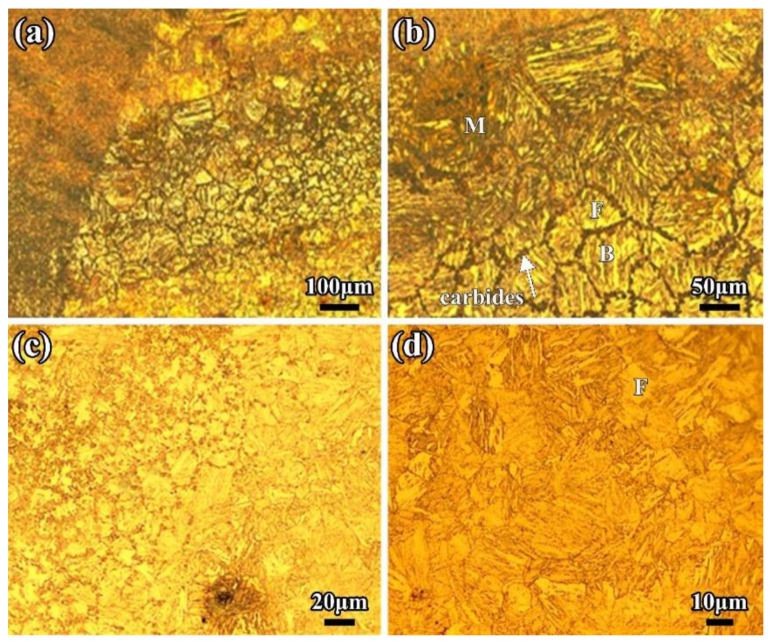
Microstructure in HAZ of SMAW joint (**a**) Macro-photo in HAZ (**b**) Chain-distribution of carbides in HAZ (**c**) OTZ (**d**) OTZ close to BM. F: ferrite; M: martensite; B: bainite.

**Figure 11 materials-16-02212-f011:**
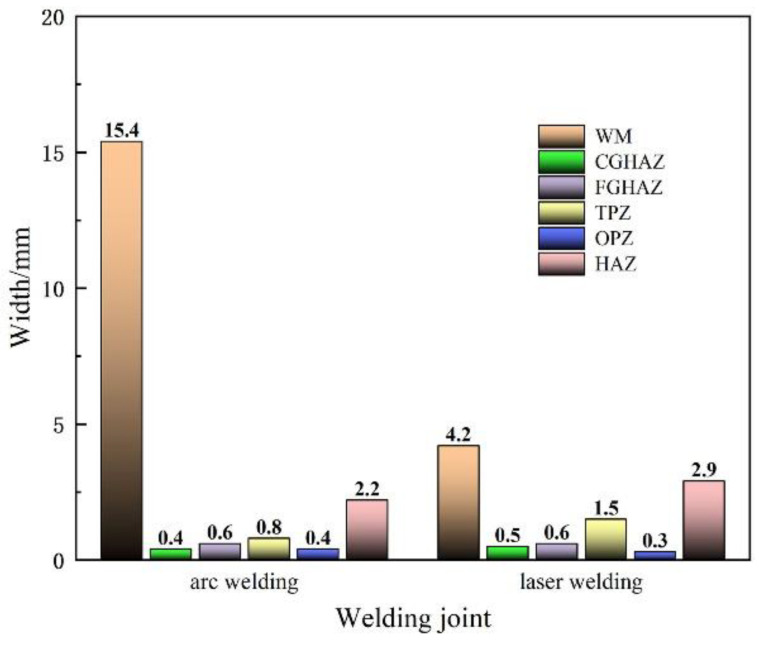
Dimensions of each zone in different welding joints.

**Figure 12 materials-16-02212-f012:**
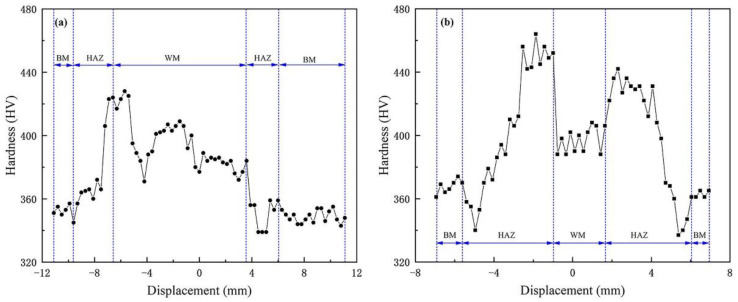
Vickers microhardness distributions (**a**) Hardness area of the SMAW joint (**b**) Hardness area of the LBW joint.

**Figure 13 materials-16-02212-f013:**
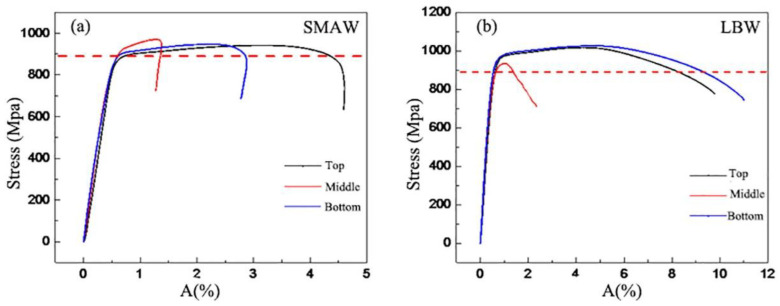
Stress–strain curves of the joints (**a**) the SMAW (**b**) the LBW.

**Figure 14 materials-16-02212-f014:**
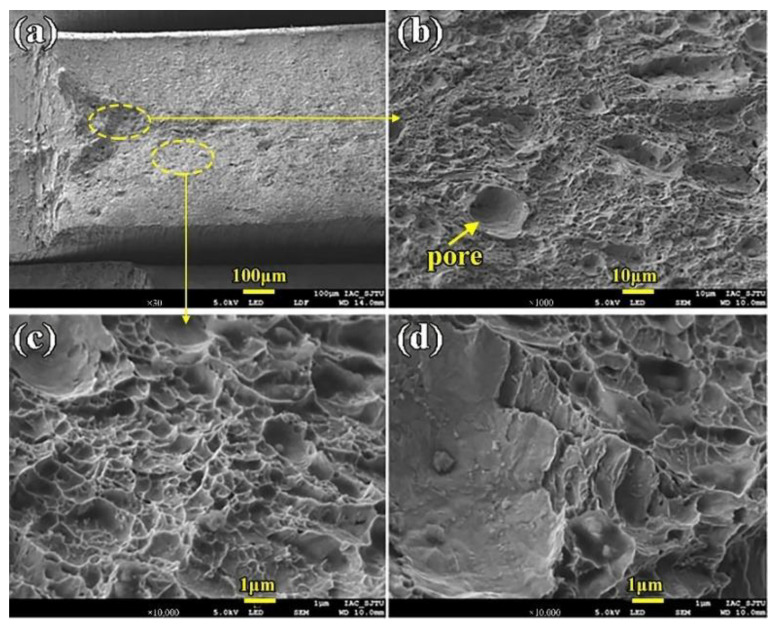
The WM impact fracture surface morphology (**a**) Macro-morphology (**b**) Fiber area morphology (**c**) Dimples details in fiber area (**d**) Dimples details in radiation area.

**Table 1 materials-16-02212-t001:** Composition and content of chemical elements in HSLA Q890 and X90 filling wire (wt.%).

	C	Si	Mn	Cr	Mo	Nb	V	Ni	W	B	P	S
Q890	0.18	0.47	1.10	0.78	0.05	0.05	0.07	0.16	0.53	0.01	0.01	0.01
X90	0.10	0.80	1.80	0.35	0.60	-	-	2.30	-	-	-	-

**Table 2 materials-16-02212-t002:** Phase content in the OTZ of the LBW joint.

Phase Content	FCC	BCC	Fe_3_C	Unidentified Resolution
Phase percentage	0.01	74.44	0.77	24.77
Phase count	3	16749	174	5574

**Table 3 materials-16-02212-t003:** Microstructure comparison between two processes.

Welded Joint	OTZ	TPZ	FGHAZ	CGHAZ	WM
LBW (9.6 kJ/cm)	multipase + polygon F	multipase + band F	Fine M + B + few F	Coarse M + B	F + B + few M
SMAW (16.4 kJ/cm)	unobvious	unobvious band	Majority B + F, few M	Majority B, few F	F + B, Complex

**Table 4 materials-16-02212-t004:** Results of tensile test.

Welding Joint	Notch Position	Impact Energy (J, −40 °C)			
		①	②	③	④
BM	BM	32.21	unbroken	unbroken	30.59
LBW	WM	15.40	3.16	17.66	16.23
	HAZ	14.92	24.10	19.82	20.87
SMAW	WM	13.46	16.96	15.84	14.37
	HAZ	20.40	19.52	18.86	31.96

**Table 5 materials-16-02212-t005:** Results of the Charpy impact test.

Sample Position		*E* (GPa)	*R*_p0.2_ (MPa)	*R*_m_ (MPa)	*A* (%)	Fracture Position
LBW	top	175	901	969	9.3	BM
	middle	189	896	935	2.0	WM
	bottom	198	895	978	10.7	BM
SMAW	top	193	900	955	1.4	BM
	middle	170	894	950	2.8	BM
	bottom	175	895	948	4.5	BM

## Data Availability

We are willing to share our research data in MDPI journals.

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
