# Peer review of "A Comparative Study on Microstructural Characterization of Thick High Strength Low Alloy Steel Weld by Arc Welding and Laser Welding"

_materials, 2023, doi:10.3390/ma16062212_

Round 1

Reviewer 1 Report

This study compares the microstructural features and mechanical qualities of joints made of 16-mm-thick high strength low alloy (HSLA) Q890 steel using single-layer autogenous laser beam welding and multi-layer, multi-pass shielded metal arc welding (SMAW) with filler wire (LBW). The reported results and the discussion are satisfactory. This manuscript is acceptable for publication with some minor changes.

Please address the following issues:

1. Define all the abbreviations before use. I noticed HAZ is not defined in the abstract. However, it is being used.

2. In the discussion section, mention what are Rp0.2(MPa), Rm(MPa), and A(%) from Table 4.

3. Lines 88-90, authors wrote:

“The absorbed energy value recorded for each joint was the average of four trials. The schematic of unstandardized 2.5*10*55 mm specimen with a V-notch is 89 presented in Figure 1(d).”

4. Please mention how the absorbed energy was calculated. Also fix the typos such 2.5*10*55. Use × instead of *. Fix such problems throughout the manuscript.

5.  Lines 286-287, authors wrote:

Meanwhile, the differences in microstructure can be measured by the content of phases and even be compared with the width of each zone in particular.

This sentence is confusing. How the microstructure of a material can be ascertained with the content of phases. Please rephrase the sentence.

6. It would be appropriate to compare the results from the current study with other findings.

7. Please explain briefly why the following steps are necessary:

Preheating temperature before welding was 120 â—¦C, the inter-pass temperature was strictly controlled below 200 â—¦C and post-weld heat treatments were adopted. After welding the weldment was cooled in air for 20 minutes, then 74 heated to 360 â—¦C in 60 minutes and kept at it for 30 minutes in heat treatment furnace 75 before being cooled in air.

Author Response

Thank you for your kind support, we carefully revised as your suggestions. The highlighting revised can be found in revised manuscript.

Reviewer 2 Report

- English language of the manuscript is acceptable in general. However, it would be much better to improve. Please avoid unnecessary long sentences. Also, some grammatical and typos mistakes can be observed: for example multipase, Mpa.

- The "Abstract" section should contain the main achievements of the research, not a general discussion. Re-organization of the abstract is needed.

- It is suggested to use more advanced characterization methods to complete the discussion regarding the difference in structure and properties in two welds.

- Lines 27 and 231: “Sorbate” should be changed to “sorbite”.

- Line 92: “grounding” should be changed to “grinding”

- The following sentence is unintelligible and should be corrected: “Figure 13(d) shows a high-magnification dimple photo in the radiative region, where the shear characteristics are more significant than those in the fiber region.”

- All parameters used in formulas must be explained. It is recommended to attach all parameters and abbreviations used in a table at the end of the article.

- Abbreviations/ acronyms, should all be defined at their first occurrence in the manuscript;

- The novelty of the work at the end of the manuscript “introduction” is not sufficient and should be explained more.

- In the "Conclusion" section, the authors should present more quantitative data as the main results of the research study rather than just some qualitative data.

- Literature review is not sufficient and authors must review and cite more papers in the field of “Modeling and predicting the relationship between properties and structure in high-strength low-alloy steels”, especially newly published ones. Doing this, reviewing the following refs could be helpful: Neural Network World 23, no. 2 (2013): 117., Materials Science and Technology 30, no. 4 (2014): 424-433, 

Author Response

Thank you for your kind support, we carefully revised as your suggestions.The highlighting revised can be found in revised manuscript.

Reviewer 3 Report

The manuscript studies the microstructural characterization of thick high strength low alloy steel weld by arc welding and laser welding. the manuscript has serious flaws and is not suitable for publication in Materials journal. The comments are as follows:

1- The introduction is not enough and should be improved. The motivation for the study is not clear. Also, the relevant references were not reviewed appropriately.

2- While it was explained that the LBW is attractive due to low heat input. Please explain the relationship between heat input, grain size and subsequently the properties of the alloy in LBW and SMAW. Please improve the discussion by citing the following reference: https://doi.org/10.1016/j.ijhydene.2022.04.260

3-  What is the meaning of the phase count in Table 2? The manuscript is not readable easily.

4- I could not find the stress-strain graph of the tensile testing. Please add them and extend the discussion.

5- The conclusion part is not complete and should be improved.

6- EBSD results are very limited. Please add more explanation and discussion about them. Also, please use the other results of EBSD for your discussion.

7- Overall, while the topic is interesting, it needs extensive improvement. 

Author Response

(The authors gave the same response as above.)

Round 2

Reviewer 2 Report

The revised manuscript could be considered for publication.

Reviewer 3 Report

The authors could satisfactorily answer the comments and concerns. Therefore, I recommend publication after another round of language checks.